Change in alcohol consumption and physical activity during the COVID-19 pandemic amongst 76 medical students

Sandell Christina christina.m.sandell@utu.fi
Saltychev Mikhail
Department of Physical and Rehabilitation Medicine, Turku University Hospital and University of Turku , Turku , Finland
Kabir Russell
Electronic publication date: 2021 Dec 9
Publication date: 2021
Volume: 9
Electronic Location ID: e12580
Received 2021 Aug 30; Accepted 2021 Nov 10
Copyright: © 2021 Sandell and Saltychev
Copyright year: 2021
Copyright holder: Sandell and Saltychev
License: This is an open access article distributed under the terms of the Creative Commons Attribution License, which permits unrestricted use, distribution, reproduction and adaptation in any medium and for any purpose provided that it is properly attributed. For attribution, the original author(s), title, publication source (PeerJ) and either DOI or URL of the article must be cited.
License URL: https://creativecommons.org/licenses/by/4.0/

Keywords: COVID-19, Alcohol, Physical activity, Modifiable risks

Funding: The authors received no funding for this work.

==============================
Objective

To investigate whether the COVID-19 pandemic has affected physical activity and alcohol consumption among medical students.

Methods

Cross-sectional survey study among 76 students in their second year of medical school. The Wilcoxon sign-rank test and Kruskal-Wallis H test were used to assess the difference between groups.

Results

Of 76 respondents, 68% were women, 66% were single and 34% were co-habiting. The median age was 21 years. Overall alcohol consumption decreased during the pandemic year by 12 g/week. Overall physical activity did not significantly change. The decrease in alcohol consumption was mostly caused by a change seen in a high tertile, change was −96 g/week. Alcohol consumption decreased more in women than in men, p = 0.0001.

Conclusions

It seems that alcohol consumption among medical students has decreased during the COVID-19 pandemic probably due to reduced social contacts and negative effect of social isolation. This decrease was seen especially among women and among students with higher alcohol consumption before the pandemic. Also, it seems that students had found their ways to remain active during the pandemic since the amount of leisure-time physical activity had not changed significantly.

Introduction

Since March 2020, societies all around the globe have started to shut down their activities due to the COVID-19 pandemic. Social distancing has played an integral part in controlling the spread of coronavirus. Many universities have switched to telecommuting and social gatherings have been restricted meaning a drastic change in the lifestyle of students (Tasso, Hisli Sahin & San Roman, 2021), affecting their overall health behaviors, and even escalating their mental health problems (Elmer, Mepham & Stadtfeld, 2020; Copeland et al., 2021; Okado et al., 2021; Charles et al., 2021). Physical exercise has been shown to be interconnected with good mental health during the COVID-19 pandemic (Pieh, Budimir & Probst, 2020; Caputo & Reichert, 2020).

Some studies have reported increased alcohol consumption among adults during the COVID-19 pandemic (Pollard, Tucker & Green, 2020; Grossman, Benjamin-Neelon & Sonnenschein, 2020). Two other studies have reported a decrease in alcohol consumption during the early stages of the pandemic (Kilian et al., 2021; Prestigiacomo et al., 2021). Prestigiacomo et al. (2021) have also found a change in drinking motives–social, enhancement and conformity drinking motives had decreased and coping motives increased. Physical activity among adults seems to have decreased during the COVID-19 pandemic (Ammar et al., 2020; Meyer et al., 2020; Caputo & Reichert, 2020). Respective previous observations among students have been inconsistent. Bertrand et al. (2021) have reported the decreased level of physical activity and increased alcohol consumption among students during the COVID-19 pandemic (Lechner et al., 2020; Bertrand et al., 2021). Also, another study reported increase in alcohol use among students at the beginning of the pandemic (Lechner et al., 2020). Some other studies have observed increased physical activity (Romero-Blanco et al., 2020; Gallego-Gómez et al., 2020) and decreased alcohol consumption among students during the COVID-19 pandemic (White et al., 2020; Jaffe et al., 2021; Bollen et al., 2021; Ryerson et al., 2021; Bonar et al., 2021). University students are known for occasional “social” excessive drinking (Carter, Obremski Brandon & Goldman, 2010), which might be the reason for decrease in alcohol consumption when social life is strictly limited. In normal circumstances students tend to maintain or increase drinking through the college years (Hultgren et al., 2019).

The objective was to investigate whether the COVID-19 pandemic has affected physical activity and alcohol consumption among medical students.

Methods

This was a cross-sectional online survey among students in their second year of medical school. In May 2021, invitations to participate in an anonymous survey through the Google Forms platform were sent by email to 150 students. Of them, 76 (51%) responded. The participants were asked about their weekly alcohol consumption and physical activity before and during the pandemic. The research was part of a pre-graduate training (CS) and no approvement by an ethical board was sought. Due to the functionality of Google Forms, no personal identifiers of the respondents (including meta-data like IP- or email addresses) were obtained. Also, no register of any kind was created.

The weekly amount of pure alcohol was calculated by multiplying a frequency (times a week) by the number of alcohol portions per time multiplied by 12. One portion, 12 g of pure alcohol, was defined as 12 cl of wine, or one beer, or 4 cl of strong liquor. The data on weekly physical activity (leisure-time and commuting) were collected from a multiple-choice chart defining four levels of vigorousness comparable to walking, brisk walking, jogging, and running. Each activity was graded as “not at all”, “less than 30 min”, “1 hour”, “2 to 3 hours”, or “4 hours or more hours”. The responses were converted into metabolic equivalent of task (MET) using a scheme shown in File S2.

Some background information on factors, which might potentially influence physical activity or alcohol consumption, was collected. Age was defined in full years at the time of response. Living arrangements were recorded as single vs. cohabiting. The participants were also asked whether they: own a dog; have small children; have had suffered an injury or illness during the pandemic, which might have affected their abilities to exercise; and are they competitive athletes–all the items were dichotomized as yes vs. no.

Statistical analysis

Due to the abnormal distribution, the results were reported as medians and interquartile ranges (IQRs), and non-parametric tests were employed. When appropriate, two-tailed p-values were reported considering p <= 0.05 significant. The Wilcoxon sign-rank test and Kruskal-Wallis H test were used to assess the difference between groups. Such variables as having a dog for a pet and being a competitive athlete were excluded from the analysis due to their rareness among the respondents. All analyses were performed using Stata/IC Statistical Software: Release 16. College Station (StataCorp LP, TX, USA).

Results

Of 76 respondents, 52 (68%) were women and 24 (32%) were men, 50 (66%) were single and 26 (34%) were co-habiting. Only five (7%) had a dog for a pet and only six (8%) were competitive athletes. The median age was 21 (IQR 22 to 23) years without a difference between genders. None had children. During the year of pandemic, 16 (21%) respondents experienced some disease or trauma.

Overall alcohol consumption decreased during the pandemic year (Table 1 and Fig. 1) by 12 (IQR −54 to 0) g/week. Instead, overall physical activity did not significantly change. As shown in Table 1, the decrease in alcohol consumption was mostly caused by a change seen in a high tertile, −96 (IQR −168 to −54) g/week, with no substantial change in low and mid tertiles. Concerning the changes in physical activity, tertiles demonstrated a regression to mean (Table 1). A Kruskal-Wallis H test was conducted to determine if changes in alcohol consumption and physical activity was different for three tertiles (low, mid, and high). A Kruskal-Wallis H test showed that there was a statistically significant difference in changes in alcohol consumption between the three groups, χ2(2) = 23.719, p = 0.0001, and in physical activity, χ2(2) = 12.013, p = 0.0025.

Figure 1 Change in alcohol consumption and physical activity before and during the COVID-19 pandemic.

Table 1 Alcohol consumption and physical activity before and during the COVID-19 pandemic grouped by tertiles.

Tertiles	n	Women,
n (%)	Co-habiting,
n (%)	Before pandemic	During pandemic	Change	
Median	IQR	p-value	Median	IQR	p-value	Median	IQR	p-value	
Alcohol consumption, g/week	
Low	28	22 (79%)	10 (35%)	12.0	0.0	24.0	0.0001	0.0	0.0	12.0	0.0001	0.0	−24.0	0.0	0.0001	
Mid	28	19 (68%)	12 (43%)	72.0	60.0	78.0	66.0	24.0	102.0	0.0	−30.0	12.0	
High	20	11 (55%)	4 (20%)	156.0	120.0	216.0	48.0	24.0	84.0	−96.0	−168.0	−54.0	
Total	76	52 (68%)	26 (34%)	60.0	24.0	108.0	–	24.0	0.0	72.0	–	−12.0	−54.0	0.0	<0.0001	
Physical activity, MET/week	
Low	26	16 (62%)	8 (31%)	22.0	13.8	24.9	0.0001	23.5	12.6	36.1	0.0003	9.8	−5.2	13.9	0.0025	
Mid	25	17 (68%)	11 (44%)	32.0	29.5	36.5	32.0	24.1	38.8	−1.6	−9.0	7.7	
High	25	19 (76%)	7 (28%)	51.5	46.6	54.9	45.8	31.1	53.6	−4.8	−22.5	0.0	
Total	76	52 (68%)	26 (34%)	31.6	24.9	46.6	–	32.2	20.0	44.9	–	0.0	−11.0	8.8	0.4487	

Alcohol consumption decreased more in women than in men, p = 0.0001 (Table 2 and Fig. 2). There was not significant difference in change in physical activity level between genders.

Figure 2 Alcohol consumption and physical activity before and during the COVID-19 pandemic grouped by gender.

Table 2 Alcohol consumption and physical activity before and during the COVID-19 pandemic grouped by gender.

Variable	Men	Women	Total	
Median	IQR	Median	IQR	Median	IQR	
Alcohol consumption, g/week	
Before pandemic	84.0	36.0	120.0	60.0	24.0	96.0	60.0	24.0	108.0	
During pandemic	48.0	12.0	72.0	12.0	0.0	66.0	24.0	0.0	72.0	
Change	0.0	−66.0	0.0	−12.0	−42.0	0.0	−12.0	−54.0	0.0	
p-value	0.0516	0.0001	<0.0001	
Physical activity, MET/week	
Before pandemic	29.2	24.5	42.7	32.4	25.3	47.2	31.6	24.9	46.6	
During pandemic	30.8	18.6	36.7	36.8	20.0	45.9	32.2	20.0	44.9	
Change	−4.1	−13.8	9.8	0.0	−8.1	8.8	0.0	−11.0	8.8	
p-value	0.157	0.9235	0.4487	

Discussion

The results showed that alcohol consumption among medical students has decreased significantly during the COVID-19 pandemic. This decrease was associated with higher alcohol consumption before the pandemic and with female gender. The amount of physical activity did not significantly change during the COVID-19 pandemic.

This study was limited by a small sample size and a response rate of 51%. There was no possibility to analyze non-respondents. The generalizability of the results might be affected by the fact that the respondents represented a narrow age range and a specific stage of their training–second-year medical students. The responses concerning the time before the COVID-19 pandemic required a year-long recall, which might affect the preciseness of responses. Physical activity and alcohol consumption of the respondents might vary during a year, which might cause difficulties when approximating health behaviors over such a long period. Overall, this study can be understood as an oversized online survey-based case series rather than a full-scale cohort investigation. However, even if limited, the study might show some trends existing amongst medical students. At least, the study might show that pandemic-related restrictions may affect some health behaviors.

The results were in line with previous reports on significantly decreased alcohol consumption among college students during a pandemic lockdown (White et al., 2020; Jaffe et al., 2021; Bollen et al., 2021; Ryerson et al., 2021; Bonar et al., 2021). Also, it has earlier been reported that alcohol consumption might decrease more substantially among heavy drinkers with social drinking motives (Bollen et al., 2021) and among those students, who had moved back in with their parents (White et al., 2020; Jaffe et al., 2021; Ryerson et al., 2021) as heavy drinking with peers had changed to drinking lighter with family (Jackson et al., 2021). Extreme drinking among students is known to be associated with social, enhancement and coping motives (White et al., 2016). Loneliness, difficulty with goal-directed behavior, COVID-19-related worry and reading COVID-19-related news have been associated with drinking to cope with the pandemic (Buckner et al., 2021; Mohr et al., 2021). There is also evidence that increased drinking during the COVID-19 pandemic has been more common among people with symptoms of anxiety and depression (Tran et al., 2020; Lechner et al., 2020). Previous reports on the changes in physical activity among students during pandemic have been inconsistent. Earlier observations on significant decrease (Bertrand et al., 2021) or increase in leisure-time physical activity (Romero-Blanco et al., 2020; Gallego-Gómez et al., 2020) could not be confirmed by the present study, which observed no significant change in physical activity among students. Also, the results contradict the reports on increasing alcohol consumption among students during the COVID-19 pandemic (Lechner et al., 2020; Bertrand et al., 2021). These differences might be explained by different age distributions, specificities of medical pre-graduate training, or other unknown confounders.

Further research among larger samples representing different training programs in longitudinal designs may provide valuable information on the changes in major modifiable risks (e.g., drinking, physical inactivity, obesity, and smoking) among university students during the COVID-19 pandemic. It might be important to assess the consistency of the changes after the pandemic is ended.

Conclusions

It seems that alcohol consumption among medical students might decrease during the COVID-19 pandemic. It can only be speculated that the reason might lay in reduced social contacts and negative effect of social isolation. This decrease was seen especially among women and among students with higher alcohol consumption before the pandemic. Also, it seems that students had found their ways to remain active during the pandemic since the amount of leisure-time physical activity had not changed significantly.

Supplemental Information

Supplemental Information 1 Data on alcohol consumption, physical activity and modifiable risks collected from participants through a survey.

Click here for additional data file.

Supplemental Information 2 Defining the level of physical activity in the survey. The responses were converted into MET units according to a following scheme.

Click here for additional data file.

Supplemental Information 3 The questionnaire which was sent to the students.

Click here for additional data file.

Supplemental Information 4 An English translation of the questionnaire sent to the sudents.

Click here for additional data file.

Additional Information and Declarations

Competing Interests

Author Contributions

Data Availability

The authors declare that they have no competing interests.

Christina Sandell conceived and designed the experiments, performed the experiments, authored or reviewed drafts of the paper, and approved the final draft.

Mikhail Saltychev conceived and designed the experiments, analyzed the data, prepared figures and/or tables, and approved the final draft.

The following information was supplied regarding data availability:

The raw data shows modifiable risks, alcohol consumption and physical activity before and during the COVID-19 pandemic reported by the participants in the Supplemental File.

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
