# Peer review of "Change in alcohol consumption and physical activity during the COVID-19 pandemic amongst 76 medical students"

_PeerJ, doi:10.7717/peerj.12580_

## Round 0.1 · original submission · Major Revisions

· Academic Editor

Major Revisions

Please address the comments from the reviewers.

Reviewer 1 ·

Basic reporting

Although the authors have tried to present an important topic, there are certain limitations in the study and these need to be improved significantly for submitting in such a reputable journal.

The introduction, methods, results, and discussion sections are too much thin for an original article. Overall, the smaller population size limits the scope of the manuscript to a wider number of readers.

Experimental design

As I have mentioned that 76 is a smaller number, I would suggest later on to circulate the survey to other Finnish universities also to make a multicenter data presentation. Even in 76 persons, the male-female ratio is not equal. The ethical committee's study approval number has to be mentioned in the methods section. This original primary research doesn't fall within Aims and Scope of the journal.

Validity of the findings

The lower number of participants arise the question of the statistical significance of the study findings and I would recommend increasing the number of participants. It would give a heterogeneous distribution of the participants, even some international students' experience would strengthen the study if available. As I mentioned the limited results and discussion section must be improved. In the later versions please include a section on "Strengths and limitations" of the study.

Reviewer 2 ·

Basic reporting

1. Title of the article is improper as it is mentioned university pre-graduate as target population. Rather it should be mentioned medical students, as there are significance difference between general university pre-graduate and medical students in terms of stress, workload and health awareness level.

2. Your introduction needs more details. At lines 45-63 different studies among students is reported regarding alcohol consumption and physical activity during COVID-19 pandemic, however, other relevant studies among medical students should be reported here additionally.

Experimental design

1. In the line 67, it is mentioned that this study was a cross-sectional cohort study. It is only a cross-sectional online survey which has to be mentioned instead.

2. Background information( at lines 86-91) which were collected during study have not been reflected in the results related to main outcome variables(alcohol consumption and physical activity).

3. Table-1 lacks the data regarding association between alcohol consumption, physical activity and men and single living arrangements.

Validity of the findings

1. No data regarding social contacts had been collected in this study, therefore, it is improper to mention in the conclusion (line 156-157) that alcohol consumption is reduced due to reduced social contacts.

2. This study lacks the data which may confound the alcohol consumption and physical activities before and during COVID-19 pandemic.

Additional comments

No comment

---

## Round 0.2 · accepted · Accept

· Academic Editor

Accept

Thanks for making all the necessary changes.